# AUTOMATING META-LEARNING BY LEARNING TO MAP TASK DISTRIBUTIONS TO INITIAL WEIGHTS

## ABSTRACT

Meta-learning recovers the Bayes-optimal learner for a particular distribution over tasks. However, meta-learning is slow and requires retraining from scratch if we modify that distribution. We present an approach that directly learns a mapping from a task distribution to the Bayes-optimal parameters of the learner (for a neural network, the initial weights of the network). We provide theoretical results identifying the optimal mapping for linear-Gaussian models and then demonstrate that hypernetworks can be used to learn this mapping from empirical data for both linear and non-linear models. This approach reduces the computational resources required to make adaptive Bayes-optimal learners: by leveraging the underlying structure of task distributions, we can meta-learn once and then quickly adapt to new settings with a single forward pass through the learned mapping.

## 1 INTRODUCTION

Acting adaptively in the world is a crucial prerequisite for artificial general intelligence. For example, a home chef robot should cook differently for an infant than for an esteemed dinner guest, even though in both cases it should obey common principles such as safe food handling. The process of learning from experience to solve a variety of related tasks is captured by idea of *meta-learning* (Schmidhuber, 1987; Bengio et al., 1991; Naik & Mammone, 1992; Caruana, 1998; Thrun & Pratt, 1998). In meta-learning, a *task distribution* encodes which tasks the learning agent is likely to encounter at test-time. At each iteration of meta-learning, a task is sampled from the task distribution and the learner adjusts its parameters to solve this task more effectively. Over the course of training, the learner acquires a set of parameters that are optimal for that distribution over tasks.

Although powerful, meta-learning is computationally expensive since it typically requires iterating over many tasks sampled from the target task distribution. It is therefore difficult to accommodate changes to the environment, since this would require retraining the learner to target a new task distribution. Motivated by this critical limitation, we ask the question: *can we meta-learn once and quickly adapt to new task distributions without retraining?*

We present an approach in which we learn a mapping directly from the parameters of the task distribution to the parameters of the learner. We focus on a version of Model-Agnostic Meta-Learning (MAML; Finn et al., 2017) applied to neural networks, where the meta-learned parameters are the initial weights of a network. In this setting, learning the mapping from distributions to weights can be done via a kind of *hypernetwork* (Ha et al., 2017). After meta-learning, transfer to new environments is trivially achieved by passing the new task parameters through the learned mapping.

Our method achieves efficient transfer to new task distributions with minimal loss in accuracy while taking nearly the same time to train as vanilla meta-learning. We additionally provide a theoretical characterization of the optimal weight initialization in a linear-Gaussian model. We show empirically that our learned hypernetwork is able to accurately predict these weights as a function of task parameters in both linear and non-linear settings. Our contributions can be summarized as follows:

- We show how the parameters of a diverse set of task distributions provide a valuable way to relate families of meta-learners.
- We identify the Bayes-optimal meta-learner in a linear-Gaussian setting, providing a closed-form expression for the optimal weights as a function of the task distribution.

- We demonstrate empirically in both linear and nonlinear settings how our approach enables efficient transfer to new task distributions with almost no loss in accuracy and with very little extra computation relative to vanilla meta-learning.

## 2 BACKGROUND

### 2.1 GRADIENT-BASED META-LEARNING

A variety of approaches have been taken to meta-learning, including learning the parameters of optimization algorithms (Schmidhuber, 1987; Bengio et al., 1992), learning metric spaces in which to apply learning (Bottou & Vapnik, 1992), and learning the hyperparameters of hierarchical Bayesian models (Baxter, 1998; Heskes, 1998). In gradient-based meta-learning, each task is assumed to be performed by a separate learner that is trained to perform that task via gradient-descent. The parameters of the learners are adjusted by optimizing their performance across the entire set of tasks.

Model-Agnostic Meta-Learning (Finn et al., 2017) is an influential gradient-based meta-learning algorithm. In MAML, the parameters of the learners are adjusted by differentiating through the gradient-based learning process. In a common application, we might have a set of learners that are neural networks, each with a set of weights $\boldsymbol{\theta}$. Each task $t$ has an associated loss $\ell_t(\boldsymbol{\theta})$. To capture the similarity between tasks, we assume that all learners are given initial weights $\boldsymbol{\theta}_0$. After applying one step of gradient descent, the weights of the learner performing task $t$ will then be $\boldsymbol{\theta}_t = \boldsymbol{\theta}_0 - \alpha \nabla \ell_t(\boldsymbol{\theta}_0)$, where $\alpha$ is a learning rate. The total loss across all tasks is $\mathcal{L} = \sum_t \ell_t(\boldsymbol{\theta}_t)$. This total loss can be differentiated with respect to $\boldsymbol{\theta}_0$, pushing the derivative through the gradient of $\ell_t$, making it possible to apply gradient descent to optimize $\boldsymbol{\theta}_0$ across tasks.

Differentiating through the gradient update can be computationally expensive, which motivated the development of an alternative gradient-based meta-learning algorithm, Reptile (Nichol et al., 2018). Like MAML, Reptile considers the effect of gradient descent to obtain task-specific weights $\boldsymbol{\theta}_t$ from a shared set of initial weights $\boldsymbol{\theta}_0$. However, rather than updating $\boldsymbol{\theta}_0$ by differentiating the total loss across tasks, Reptile uses each set of task-specific weights to compute a gradient $\boldsymbol{\theta}_t - \boldsymbol{\theta}_0$ that is supplied to a generic optimizer such as Adam. In practice, this approach can yield comparable results to MAML at far lower computational cost.

### 2.2 META-LEARNING AS HIERARCHICAL BAYES

As noted above, meta-learning has deep connections to hierarchical Bayesian inference (Baxter, 1998; Heskes, 1998). In a hierarchical Bayesian model, each task involves inferring some unknown parameters $\boldsymbol{\theta}_t$ based on observed data. This is done by Bayesian inference, with a prior distribution with hyperparameters $\boldsymbol{\xi}$ shared across all tasks. Those hyperparameters can be adjusted based on the information provided by the data aggregated across all tasks, providing a way to adapt the task-specific Bayesian learners based on the shared statistical structure of those tasks.

This view of meta-learning can be connected directly to the gradient-based approach outlined above (Grant et al., 2018). Gradient descent with a limited number of iterations is equivalent to finding the maximum a posteriori (MAP) parameter value with a specific Gaussian prior for linear models, with the mean of that prior corresponding to the initial weights used in gradient descent. Learning those initial weights $\boldsymbol{\theta}_0$ can thus be viewed as estimating the hyperparameters of the prior $\boldsymbol{\xi}$.

Viewed from this perspective, Reptile is also an effective algorithm for approximating hierarchical Bayesian inference in a specific class of models. In hierarchical Bayesian inference, we are interested in maximizing the marginal likelihood, which for a given task is

$$p(\boldsymbol{y}_t|\boldsymbol{\theta}_0) = \int p(\boldsymbol{y}|\boldsymbol{\theta}_t)p(\boldsymbol{\theta}_t|\boldsymbol{\theta}_0)\,d\boldsymbol{\theta}_t \tag{1}$$

$$\approx p(\boldsymbol{y}|\hat{\boldsymbol{\theta}}_t)p(\hat{\boldsymbol{\theta}}_t|\boldsymbol{\theta}_0) \tag{2}$$

where $\hat{\boldsymbol{\theta}}_t$ is the MAP value of $\boldsymbol{\theta}_t$. Taking logarithms, the term that contains $\boldsymbol{\theta}_t$ is $\log p(\hat{\boldsymbol{\theta}}_t|\boldsymbol{\theta}_0)$. Collecting these terms across tasks and seeking to minimize the negative log probability, we obtain the loss $-\sum_t \log p(\hat{\boldsymbol{\theta}}_t|\boldsymbol{\theta}_0)$. If the prior $p(\boldsymbol{\theta}_t|\boldsymbol{\theta}_0)$ is Gaussian, then this loss is proportional to $\sum_t (\boldsymbol{\theta}_t - \boldsymbol{\theta}_0)^2$, which is the implicit loss used in Reptile. Reptile thus approximately maximizes the marginal likelihood if $\boldsymbol{\theta}_t$ is assumed to follow a Gaussian prior.

Table 1: Representative examples of meta-learning problems. Each meta-learning problem is expressed through a distribution over task vectors $\mathbf{w}$ and a distribution over model parameters $\boldsymbol{\theta}$.

| Meta-learning Problem | Task Vector ($\mathbf{w}$) | Model Parameters ($\boldsymbol{\theta}$) |
|---|---|---|
| Regression | Target function | Regressor weights |
| Image classification | Target classes | Classifier weights |
| Meta-reinforcement learning | RL environment parameters | Policy network weights |
| LLM post-training | Relative weighting of data sources | Post-trained LLM weights |

### 2.3 HYPERNETWORKS

Given the high computational cost of training a neural network via gradient descent, it is attractive to find other ways to estimate the weights that should be used to perform a specific task. Intuitively, there should exist some mapping from tasks to the ideal weights of a neural network. If we could just learn that mapping, we wouldn't need to train networks via gradient descent.

Hypernetworks (Ha et al., 2017) are a method for solving this problem, using neural networks to learn the weights that should be used by other neural networks. While originally used to automatically generate blocks of weights for structured architectures such as convolutional neural networks and long short-term memory networks, hypernetworks have been used in a wide range of settings where the weights of a network need to be generated based on context (Chauhan et al., 2024).

## 3 LEARNING TO MAP TASK DISTRIBUTIONS TO INITIAL WEIGHTS

In meta-learning, task distributions are typically considered in isolation. However, by considering the relationships between task distributions we open the door to learning the corresponding relationships between meta-learners optimized under these task distributions. This way of thinking leads to new efficient methods for fine-tuning and adaptation.

In Section 3.1, we describe our meta-learning formulation, which is grounded in the principles of probabilistic inference. To demonstrate the validity of our approach, it is important to thoroughly analyze its behavior on a well-behaved problem. We therefore consider a meta-learned Bayesian linear regression model in Section 3.2 where both the task distribution and prior on model parameters are Gaussian. We derive the optimal model hyperparameters as a function of the task distribution in closed form (Theorem 3.1). Motivated by these insights, in Section 3.3 we present our practical approach for learning the mapping from task distributions to initial model weights. Our approach is a natural generalization of the hierarchical Bayesian view of meta-learning discussed in Section 2.2.

### 3.1 META-LEARNING FORMULATION

We begin by introducing a common formulation of the meta-learning problem. Our goal is exposit a framework that is sufficiently abstract to capture a wide range of problems, yet detailed enough to contextualize our novel hypernetwork-based approach in Section 3.3. Please refer to Table 1 for several representative meta-learning problems that can be expressed with this formulation.

**Task distribution.** In meta-learning, the task distribution specifies the relative importance of tasks within the environment. We assume that each task is represented by a vector $\mathbf{w}$. The task distribution therefore corresponds to some distribution $p(\mathbf{w})$ over task vectors. A task vector $\mathbf{w}$ can be used to construct a distribution over data $(\mathbf{x}, \mathbf{y})$ as follows. Let $p(\mathbf{x})$ be a distribution over inputs. Targets $\mathbf{y}$ are then sampled from a conditional distribution given the input $\mathbf{x}$ and the task vector:

$$\mathbf{x} \sim p(\mathbf{x}), \qquad \mathbf{y} \sim p(\mathbf{y} \mid \mathbf{x}, \mathbf{w}). \tag{3}$$

**Meta-learned model.** The per-task adapted parameters of the meta-learned model are denoted by $\boldsymbol{\theta}$. The model then learns a vector $\boldsymbol{\xi}$ such the distribution over task-specific parameters $p(\boldsymbol{\theta} \mid \boldsymbol{\xi})$ yields high marginal likelihood for data sampled from the task distribution. The functional form of

the learned prior distribution $p(\boldsymbol{\theta} \mid \boldsymbol{\xi})$ depends on the method. For MAML, $\boldsymbol{\xi}$ is simply a weight initialization $\boldsymbol{\theta}_0$ and the distribution over weights is defined implicitly centered at $\boldsymbol{\theta} = \boldsymbol{\theta}_0$.

**Meta-learning objective.** The objective of the meta-learning is to minimize the negative log marginal likelihood using data sampled from the task distribution:

$$\mathcal{L}(\boldsymbol{\xi}; \mathbf{y}) \triangleq p(\mathbf{y} \mid \boldsymbol{\xi}) = \int p(\mathbf{y} \mid \boldsymbol{\theta}) p(\boldsymbol{\theta} \mid \boldsymbol{\xi}) \, d\boldsymbol{\theta} \tag{4}$$

$$J(\boldsymbol{\xi}; \mathbf{y}) \triangleq - \log \mathcal{L}(\boldsymbol{\xi}; \mathbf{y}). \tag{5}$$

Meta-learning with a marginal likelihood objective is a promising approach since it recovers Bayes-optimal learners relative to the task distribution (Aitchison, 1975; Mikulik et al., 2020; Binz et al., 2024). Other learning objectives are possible, including the predictive likelihood computed over held-out data within each task. These two objectives differ in their emphasis on modeling per-task data: marginal likelihood focuses on all data within the task whereas predictive likelihood focuses on the held-out data alone. The relative merits has been discussed in previous work by Snell & Zemel (2021). For the sake of conceptual simplicity, we focus here on marginal likelihood.

### 3.2 OPTIMAL META-LEARNING IN A TRACTABLE SETTING: BAYESIAN LINEAR REGRESSION

In this section, we consider a worked example where the learner is a Bayesian linear regression model. Both the task vectors and model parameters are generated from a Gaussian distribution. Each task vector represents a target function which is potentially nonlinear. The model predictions are taken to be linear. These assumptions allow us to derive the optimal model hyperparameters as a function of the task hyperparameters. This relationship between the task hyperparameters and optimal model hyperparameters will be useful to motivate our method in Section 3.3.

**Task distribution.** The task distribution is specified as follows:

$$\mathbf{w} \sim \mathcal{N}(\boldsymbol{\mu}_{\mathbf{w}}, \boldsymbol{\Sigma}_{\mathbf{w}}), \qquad \mathbf{x}_i \sim \mathcal{N}(\boldsymbol{\mu}_{\mathbf{x}}, \sigma_{\mathbf{x}}^2 \mathbf{I}) \text{ for } i = 1, 2, \ldots, n, \qquad \mathbf{y} \sim \mathcal{N}(\boldsymbol{\Phi}\mathbf{w}, \sigma^2 \mathbf{I}), \tag{6}$$

where $\boldsymbol{\eta} \triangleq \{\boldsymbol{\mu}_{\mathbf{w}}, \boldsymbol{\Sigma}_{\mathbf{w}}\}$ are the task hyperparameters, $\mathbf{x}_i$ is an input example, and $\boldsymbol{\Phi} \in \mathbb{R}^{n \times p}$ is a feature matrix where each row is obtained by applying an arbitrary (possibly nonlinear) feature function $\phi$ to to the corresponding input $\mathbf{x}_i$. Note that the target function for a task may be expressed explicitly as $f_{\mathbf{w}}(\mathbf{x}) = \langle \phi(\mathbf{x}), \mathbf{w} \rangle$

**Meta-learning model.** We assume the model is standard Bayesian linear regression. The predictive distribution over a vector of regression targets $\mathbf{y} \in \mathbb{R}^n$ is a Gaussian with the predictive mean being a linear function of the inputs:

$$p(\mathbf{y} \mid \boldsymbol{\theta}) = \mathcal{N}(\mathbf{y} \mid \mathbf{X}\boldsymbol{\theta}, \sigma_0^2 \mathbf{I}), \tag{7}$$

where $\mathbf{X} \in \mathbb{R}^{n \times d}$ is the design matrix obtained by taking each row to be an input example $\mathbf{x}_i$. Let the model hyperparameters $\boldsymbol{\xi} \triangleq \{\mathbf{m}_0, \mathbf{P}_0\}$ consist of a prior mean $\mathbf{m}_0 \in \mathbb{R}$ and a prior covariance $\mathbf{P}_0 \in \mathbb{R}^{d \times d}$. Similar to the predictive distribution, the model distribution over $\boldsymbol{\theta}$ is also assumed to be Gaussian:

$$p(\boldsymbol{\theta} \mid \boldsymbol{\xi}) = \mathcal{N}(\boldsymbol{\theta} \mid \mathbf{m}_0, \mathbf{P}_0), \tag{8}$$

This conjugacy between the predictive distribution the and prior allows the marginal likelihood objective to be computed in closed-form, as we shall soon see.

**Marginal Likelihood.** The marginal likelihood given observed task data is obtained by marginalizing over the model parameters $\boldsymbol{\theta}$:

$$\mathcal{L}(\mathbf{m}_0, \mathbf{P}_0; \mathbf{y}) \triangleq p(\mathbf{y} \mid \mathbf{m}_0, \mathbf{P}_0) = \int \mathcal{N}(\mathbf{y} \mid \mathbf{X}\boldsymbol{\theta}, \sigma_0^2 \mathbf{I}) \mathcal{N}(\boldsymbol{\theta} \mid \mathbf{m}_0, \mathbf{P}_0) \, d\boldsymbol{\theta} \tag{9}$$

$$= \mathcal{N}(\mathbf{y} \mid \mathbf{X}\mathbf{m}_0, \mathbf{X}\mathbf{P}_0\mathbf{X}^\top + \sigma_0^2 \mathbf{I}).$$

We can further marginalize over $\mathbf{y}$ (Equation 6) to compute the expected marginal likelihood given the task vector $\mathbf{w}$:

$$\mathcal{L}(\mathbf{m}_0, \mathbf{P}_0; \mathbf{w}) = \int \mathcal{N}(\mathbf{y} \mid \boldsymbol{\Phi}\mathbf{w}, \sigma^2 \mathbf{I}) \mathcal{L}(\mathbf{m}_0, \mathbf{P}_0; \mathbf{y}) \, d\mathbf{y}$$

$$= \mathcal{N}(\mathbf{X}\mathbf{m}_0 \mid \boldsymbol{\Phi}\mathbf{w}, \mathbf{X}\mathbf{P}_0\mathbf{X}^\top + (\sigma_0^2 + \sigma^2)\mathbf{I}).$$

Moreover, using the fact that $\mathbf{w} \sim \mathcal{N}(\boldsymbol{\mu}_{\mathbf{w}}, \boldsymbol{\Sigma}_{\mathbf{w}})$, the task vectors can be marginalized over as well:

$$\mathcal{L}(\mathbf{m}_0, \mathbf{P}_0) = \int p(\mathbf{w} \mid \boldsymbol{\mu}_{\mathbf{w}}, \boldsymbol{\Sigma}_{\mathbf{w}}) \mathcal{L}(\mathbf{m}_0, \mathbf{P}_0; \mathbf{w}) \, d\mathbf{w} \tag{10}$$

$$= \mathcal{N}(\mathbf{X}\mathbf{m}_0 \mid \boldsymbol{\Phi}\boldsymbol{\mu}_{\mathbf{w}}, \mathbf{X}\mathbf{P}_0\mathbf{X}^\top + \boldsymbol{\Phi}\boldsymbol{\Sigma}_{\mathbf{w}}\boldsymbol{\Phi}^\top + (\sigma_0^2 + \sigma^2)\mathbf{I}).$$

Now we are ready to solve for the model hyperparameters $\boldsymbol{\xi}^*$ that minimize the negative log marginal likelihood $J(\mathbf{m}_0, \mathbf{P}_0) \triangleq -\log \mathcal{L}(\mathbf{m}_0, \mathbf{P}_0\mathbf{X})$.

**Theorem 3.1** (Optimal model hyperparameters for meta-learned Bayesian linear regression)**.** *Let the task distribution and model distribution be defined as in Section 3.2. Then the model hyperparameters that minimize the negative log marginal likelihood $J(\mathbf{m}_0, \mathbf{P}_0)$ are given by:*

$$\mathbf{m}_0^* = (\mathbf{X}^\top\mathbf{X})^{-1}\mathbf{X}^\top\boldsymbol{\Phi}\boldsymbol{\mu}_{\mathbf{w}} \tag{11}$$

$$\mathbf{P}_0^* = (\mathbf{X}^\top\mathbf{X})^{-1}\mathbf{X}^\top\left[\mathbf{r}\mathbf{r}^\top - \left(\boldsymbol{\Phi}\boldsymbol{\Sigma}_{\mathbf{w}}\boldsymbol{\Phi}^\top + (\sigma_0^2 + \sigma^2)\mathbf{I}\right)\right]\mathbf{X}(\mathbf{X}^\top\mathbf{X})^{-1}, \tag{12}$$

*where $\mathbf{r} = (\mathbf{X}\mathbf{m}_0^* - \boldsymbol{\Phi}\boldsymbol{\mu}_{\mathbf{w}})$.*

*Proof.* Proofs for all theoretical results may be found in Appendix A. $\square$

Theorem 3.1 shows how the optimal hyperparameters $\boldsymbol{\xi}^* = \{\mathbf{m}_0^*, \mathbf{P}_0^*\}$ can be expressed in terms of the task distribution parameters $\boldsymbol{\eta} = \{\boldsymbol{\mu}_{\mathbf{w}}, \boldsymbol{\Sigma}_{\mathbf{w}}\}$. We exploit this observation in the next section to derive our hypernetwork-based approach.

### 3.3 APPROXIMATELY OPTIMAL META-LEARNING VIA HYPERNETWORKS

Recall in the previous section how we showed that the optimal model hyperparameters $\boldsymbol{\xi}^*$ can be expressed as a function of the task distribution parameters $\boldsymbol{\eta}$ in the case of meta-learned Bayesian linear regression. In more general meta-learning problems, we may not be able to derive a closed-form expression for the optimal hyperparameters, as we did in Theorem 3.1. Therefore, in this section we discuss our proposed approach to *learn* the mapping $\boldsymbol{\eta} \to \boldsymbol{\xi}$ from the task distribution parameters to model hyperparameters.

In the general meta-learning problem, we assume that the underlying model is a deep neural network. The model generates a predictive distribution over targets by passing the input data through a deep neural network with parameters $\boldsymbol{\theta}$. This setup captures modeling in many prominent domains, including classification, regression, autoregressive language modeling, and reinforcement learning.

A naïve approach to modeling $p(\boldsymbol{\theta} \mid \boldsymbol{\xi})$ would be computationally costly due to the need to perform probabilistic inference over the weights of a neural network. To derive a more practically feasible algorithm, we let the model hyperparameters be simply the initialization $\boldsymbol{\theta}_0$ of the neural network weights. These weights are adapted to each task by a few steps of gradient descent, as in MAML.

Since the output of the learned mapping $\boldsymbol{\eta} \to \boldsymbol{\xi}$ is therefore a set of neural network weights $\boldsymbol{\theta}_0$, we propose to let the mapping be represented by a hypernetwork with parameters $\phi$:

$$\boldsymbol{\theta}_0 \leftarrow h_\phi(\boldsymbol{\eta}) \tag{13}$$

The parameters of the hypernetwork are meta-learned in a manner analogous to Reptile:

$$\mathcal{L}(\phi) = \|h_\phi(\boldsymbol{\eta}) - \hat{\boldsymbol{\theta}}\|^2, \tag{14}$$

where $\hat{\boldsymbol{\theta}} = \boldsymbol{\theta}_0 - \alpha\nabla_\theta J$ are the model parameters after adaptation to the task. This objective is simple and computationally expedient, yet remains connected to the original goal of maximizing marginal likelihood since Reptile can be viewed as approximately maximizing marginal likelihood assuming an isotropic distribution in weight space. Other meta-learning objectives are possible here but we focus on the learning rule from Equation 14 due to its effectiveness and conceptual simplicity.

Over the course of meta-learning, the hypernetwork learns how to predict good parameter initializations for solving tasks sampled from the task distribution. Note that here, unlike the previous section, we do not assume knowledge of the task distribution's functional form. We only need to be able to sample task data in order to be able to minimize Equation 14.

The key benefit of our approach is the ability to seamlessly adapt to new task distributions. Since each task distribution is represented by parameters $\boldsymbol{\eta}$, we can easily adapt to a new task distribution (corresponding to a new task environment) with parameters $\tilde{\boldsymbol{\eta}}$ by a single forward pass through the hypernetwork to get a new predicted weight initialization $\tilde{\theta}_0 = h_{\boldsymbol{\phi}}(\tilde{\boldsymbol{\eta}})$. This highlights the benefits of meta-learning once with a diverse set of task distributions: we no longer need to undergo the expensive meta-learning process again upon change of the environment.

## 4 EXPERIMENTS

In this section we examine the experimental performance of our proposed method. In Section 4.1, we confirm the validity of our theoretical result for the optimal model hyperparameters in the meta-learned Bayesian linear regression setting. Then we conduct experiments that show the benefits of our hypernetwork-based approach relative to standard meta-learning baselines in both linear-Gaussian (Section 4.2) and nonlinear MLP (Section 4.3) domains.

### 4.1 VERIFICATION OF OPTIMAL META-LEARNED HYPERPARAMETERS

In this section we empirically validate our analytical solution for the optimal mean hyperparameter $\mathbf{m}_0^*$ from Equation 11. We ran two experiments to verify that our analytically derived expression coincides with the empiricially derived minimizer of the negative log marginal likelihood. We ran two experiments in $d = 2$ dimensions with $n = 50$ samples each. In both cases the input examples were drawn i.i.d. from a standard normal $\mathbf{x}_i \sim \mathcal{N}(\mathbf{0}, \mathbf{I})$ and we fixed $\mathbf{w} = \boldsymbol{\mu}_{\mathbf{w}} = [2.0, -1.0]^\top$. We compared two sets of features: $\phi(\mathbf{x}) = \mathbf{x}$ and $\phi(\mathbf{x}) = \mathbf{A}\mathbf{x}$ for fixed $\mathbf{A} = \begin{bmatrix} 0.5 & 1 \\ 1.5 & 1 \end{bmatrix}$.

For each case we computed the closed-form expression $\mathbf{m}_0^* = (\mathbf{X}^\top \mathbf{X})^{-1} \mathbf{X}^\top \boldsymbol{\Phi} \boldsymbol{\mu}_{\mathbf{w}}$ and compared this to 50 steps of gradient descent initialized at $\mathbf{m}_0^{(0)} = [-2, -2]^\top$ with learning rate $\alpha = 0.01$:

$$\mathbf{m}_0^{(t)} \leftarrow \mathbf{m}_0^{(t-1)} - \alpha \nabla_{\mathbf{m}_0} J(\mathbf{m}_0^{(t)}) \text{ for } t = 1, 2, \ldots, 50. \tag{15}$$

Figure 1 shows the contour plots of the loss $J(\mathbf{m}_0) = \frac{1}{2} \|\mathbf{X}\mathbf{m}_0 - \phi(\mathbf{X})\mathbf{w}\|^2$, the gradient-descent trajectory, and the optimal analytic solution $\mathbf{m}_0^*$. These results demonstrate that our closed-form solution does indeed minimize $J(\mathbf{m}_0)$.

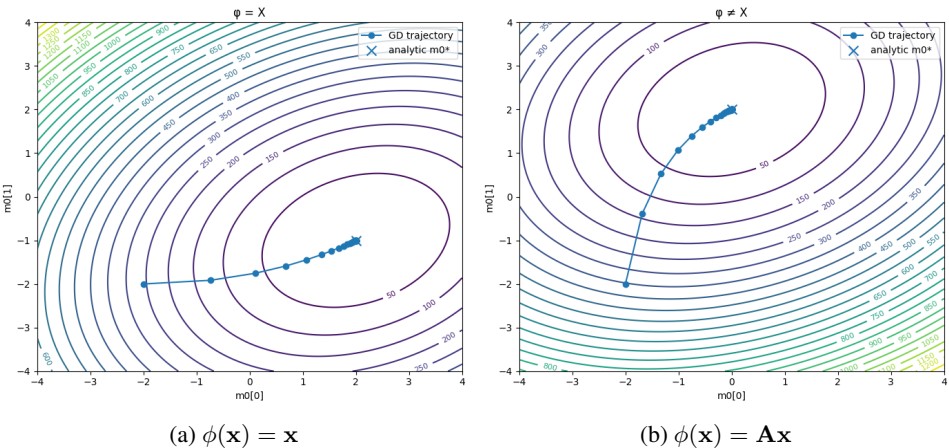

(a) $\phi(\mathbf{x}) = \mathbf{x}$            (b) $\phi(\mathbf{x}) = \mathbf{A}\mathbf{x}$

Figure 1: Contour of $J(\mathbf{m}_0)$ and gradient descent path for two choices of $\phi(\mathbf{x})$. In both cases, the gradient descent trajectory converges to the analytic expression for $\mathbf{m}_0^*$.

### 4.2 LINEAR-GAUSSIAN EXPERIMENTS

Table 2: Comparison of meta-learning approaches on Gaussian linear regression. Higher $R^2$ and lower nMSE are better. Pooled MAML shares an initialization $\boldsymbol{\theta}_0$ across all task distributions while Oracle MAML is able to retrain a separate $\boldsymbol{\theta}_0$ for each.

| Method | $R^2$ ($\uparrow$) | nMSE ($\downarrow$) |
|---|---|---|
| Oracle MAML (per-prior re-trained) | 0.8111 | 0.7062 |
| Pooled MAML (shared initialization) | 0.1259 | 0.8557 |
| Meta-learned Hypernetwork (Ours) | **0.8111** | **0.7045** |

**Setup.** We now empirically validate our proposed hypernetwork approach in a linear-Gaussian setting. As in previous sections, we consider a regression task where each task vector represents a target function $f_{\mathbf{w}}(\mathbf{x}) = \mathbf{x}^\top \mathbf{w}$. Input points $\mathbf{x}_i \in \mathbb{R}^2$ are drawn i.i.d. from a standard Gaussian $\mathbf{x}_i \sim \mathcal{N}(\mathbf{0}, \mathbf{I})$ and the targets are sampled as $y_i \sim \mathcal{N}(f_{\mathbf{w}}(\mathbf{x}_i), \sigma_y^2)$. Task vectors are sampled as $\mathbf{w}_i \sim \mathcal{N}(\boldsymbol{\mu}_{\mathbf{w}}, \sigma_{\mathbf{w}}^2 \mathbf{I})$, where $\boldsymbol{\mu}_{\mathbf{w}} \in \mathbb{R}^2$. Each task consists of a *support set* of labeled data for task-specific adaptation and a *query set* on which predictive performance is computed.

**Hypernetwork learning.** The goal of standard meta-learning in this context is to learn an initialization $\boldsymbol{\theta}_0 \in \mathbb{R}^2$ such that one gradient descent step on the new task yields model parameters with low mean-squared error on the targets. In contrast, our hypernetwork-based meta-learning algorithm learns to directly predict a good initialization through the mapping $h_\phi$:

$$\boldsymbol{\theta}_0 = h_\phi(\boldsymbol{\mu}_{\mathbf{w}}). \tag{16}$$

In order to meta-learn with a diversity of task distributions, $\boldsymbol{\mu}_{\mathbf{w}}$ is sampled uniformly at random from $[-3, 3]^2$ over the course of meta-learning. Given $\boldsymbol{\mu}_{\mathbf{w}}$, we sample many tasks $\mathbf{w}_i$ and for each task perform one-step adaptation on the support set, initialized at $\boldsymbol{\theta}_0$. After adaptation, loss is computed on the query set. Then the hypernetwork parameters are updated by backpropagation in order to minimize this query loss by taking a gradient descent step with respect to the hypernetwork parameters $\phi$. This procedure enables generalization to new task distributions without re-training.

**Baselines.** We compare the results of the meta-learned hypernetwork to two baselines. The first, *Pooled MAML*, is equivalent to our method but has no hypernetwork. Instead, a single setting of $\boldsymbol{\theta}_0$ is learned that is shared across all samples of $\boldsymbol{\mu}_{\mathbf{w}}$. This baseline measures the increase in performance due to the hypernetwork. We also compare to a separate oracle baseline, *Oracle MAML*, which has oracle access to the test-time priors and meta-learns a separate setting of $\boldsymbol{\theta}_0$ for each prior by sampling task vectors directly. This baseline represents an upper bound on how well we can reasonably expect our hypernetwork-based method to perform. However, this level of performance is typically unachievable in practice because we rarely have privileged access to test-time task distributions beforehand. Details of the settings for all algorithms are in Appendix B.2.

**Results.** For all methods, we evaluate performance after one inner gradient update on each task's support set, and report the coefficient of determination ($R^2$) and normalized mean squared error (nMSE) on a held-out test prior using a larger evaluation set. Results are shown in Table 2. Our meta-learned hypernetwork achieved the same performance as Oracle MAML, recovering the optimal solution, and significantly outperformed Pooled MAML.

### 4.3 META-LEARNED MULTILAYER PERCEPTRONS

**Setup.** Our results so far have confirmed the Bayes-optimal mapping in the linear case and demonstrated that this mapping can be learned from data. We now show that the same approach can be applied in a non-linear setting, demonstrating that it is possible to learn a hypernetwork that identifies the optimal initial weights for multilayer perceptrons (MLPs).

Let $f_\theta : \mathbb{R}^2 \to \mathbb{R}$ be a two-layer MLP with parameter vector $\theta \in \mathbb{R}^P$. We represent each task distribution by a 2-vector $\mu_w \in \mathbb{R}^2$. For each task distribution $\mu_w$, we define a mean in **parameter**

Table 3: Neural network case: performance on a held-out test prior after $K=3$ inner updates. Higher $R^2$ and lower nMSE are better.

| Method | $R^2$ ($\uparrow$) | nMSE ($\downarrow$) |
|---|---|---|
| Oracle MAML (per-prior re-trained) | 0.6801 | 0.3183 |
| Pooled MAML (shared init) | 0.6523 | 0.3460 |
| Meta-learned Hypernetwork (Ours) | **0.6662** | **0.3321** |

**space** via a fixed mapping $m : \mathbb{R}^2 \to \mathbb{R}^P$: $m(\mu_w) \in \mathbb{R}^P$. Task-level parameters $w_i \in \mathbb{R}^P$ are then drawn from a Gaussian around that mean:

$$w_i \sim N\big(m(\mu_w), \sigma_w^2 I_P\big).$$

In our experiment we take $\mu_w$ to be uniformly distributed over $[-3, 3]^2$.

For each task, the inputs $x \in \mathbb{R}^2$ are i.i.d. standard Gaussian,

$$x \sim N(0, I).$$

and the target outputs are produced by a MLP with the task's weights, plus observation noise,

$$y = f_{w_i}(x) + \varepsilon, \quad \varepsilon \sim N(0, \sigma_y^2).$$

We then generate a **support set**: $\{(x_s, y_s)\}_{s=1}^{k_{\mathrm{spt}}}$ and a **query set**: $\{(x_q, y_q)\}_{q=1}^{k_{\mathrm{qry}}}$ for that task.

**Hypernetwork learning**   We seek a set of initial weights $\theta_{\mathrm{init}} \in \mathbb{R}^P$ such that one gradient step on a new task's support set yields parameters $\theta_i$ that have low mean-squared error on that task's query set. Rather than running MAML's outer loop each time we see a new prior $\mu_w$, we train a hyper-network:

$$h_\phi : \mathbb{R}^2 \to \mathbb{R}^P$$

to predict an initialization directly from $\mu_w$:

$$\theta_{\mathrm{init}} = h_\phi(\mu_w)$$

that yields low post-adaptation loss. As in the previous section, this hypernetwork is trained by backpropagating through the task loss.

At test time, for a brand-new prior $\mu_w$, we only need a single forward pass through $h_\phi$ to obtain $\theta_{\mathrm{init}}$ — no task sampling or inner meta-loops.

**Baselines**   We compare to two approaches. First, *MAML (per-prior re-trained)* learns a separate initialization for each test prior. Second, *Pooled MAML* uses a single shared initialization (and shared step-size parameters) across all priors.

For all methods we evaluate after $K = 3$ *inner gradient updates* on each task's support set and report coefficient of determination ($R^2$) and normalized mean squared error (nMSE) on a held-out test prior using a larger evaluation set. Details of the settings for all algorithms are in Appendix B.3.

**Results**   The results are given in Table 3. Our hypernetwork approach achieves performance that is close to Oracle MAML, despite significantly less computational cost since it does not need to re-train for each prior. It also gets closer to the oracle results than Pooled MAML, demonstrating that it is able to adapt the solution it finds to the different priors appropriately.

## 5   RELATED WORK

The idea of predicting the weights of a neural network as a function of context has been used in a variety of settings (Ha et al., 2017; Chauhan et al., 2024). We focus here on its use in meta-learning and reinforcement learning, as these are two prime cases where the behavior of the network should

change across episodes. This is due to the inherent stochasticity in these settings – over tasks for meta-learning or over the environment for reinforcement learning.

Meta-learning can be applied to a wide range of problems. One of the most relevant problems is zero-shot learning in which a task description vector or other side information such as text is provided to the base learner. Early approaches to this task uses textual descriptions of novel classes to predict the weights of the classification layer of a convolutional neural network (Ba et al., 2015). Similar techniques have been applied in meta-learning solutions to the zero-shot problem, for example by learning to predict class prototypes from side information (Snell et al., 2017).

The hierarchical Bayesian perspective on meta-learning (Grant et al., 2018) offers a different way of thinking about this problem, in which the goal is not simply to learn a single weight initialization but rather a distribution over weights that assigns high probability to the optimal weights across different tasks. This can be viewed as learning a generative model for the learner's weights. In the spirit of variational autoencoders (Kingma & Welling, 2019), this suggests an amortized inference approach in which the approximate posterior over weights can be predicted as a function of the labeled examples. Such an approach was explored by Gordon et al. (2019). Interestingly, learning the conditional mean alone (i.e., ignoring the variance) can itself be viewed as a form of hypernetwork learning. Qiao et al. (2018) develop an approach along these lines and apply it to few-shot classification. Hypernetworks have also been used to make meta-learning more efficient by reducing the dimensionality of the learning problem (Zhao et al., 2020). More recently, similar methods have been used to learn a distribution over the parameters of LoRA adapters (Hu et al., 2021) to support fast LLM adaptation (Zhang et al., 2025). Unlike previous methods, which apply amortized inference inside the inner adaptation loop, ours uses hypernetworks at a higher level: from the task distribution parameters to the meta-level parameters directly.

Similar concerns arise in other problem domains. Zero-shot transfer is a key challenge in reinforcement learning. Hypernetworks have been applied to this problem in previous work (HyperZero; Rezaei-Shoshtari et al., 2023). In HyperZero, policies are first independently learned for parameterized Markov Decision Processes across different parameter settings. Then a hypernetwork is learned in a supervised manner to predict the weights of the policy network as a function of MDP parameters. In contrast to ours, this approach is not end-to-end but decouples the hypernetwork learning into two stages: learning many different policy networks, one for each MDP, and then consolidating them by learning a supervised hypernetwork.

Another related approach comes from the literature on physics-informed neural networks (PINNs). One primary application of PINNs is to solve differential equations using neural networks. Here it is often advantageous to be solve different problems with the same functional form where the parameters may change. To attack this setting, an approach called solution bundles (Flamant et al., 2020; Flores et al., 2025) have been proposed, where the weights of the neural networks are learned as a function of the differential equation parameters.

To our knowledge, ours is the first to unify these related approaches within the meta-learning paradigm while maintaining a focus on learning a mapping from task distributions to initial model weights. This approach improves adaptability to new task environments by removing the need to retrain the meta-learner from scratch.

## 6 CONCLUSION

Creating artificial general intelligence systems that are able to rapidly adapt their behavior to a wide range of settings requires instilling effective inductive biases in those systems. Meta-learning is a promising approach for doing so, creating systems that have initial weights that instantiate a specific prior distribution over tasks and can be easily adapted with additional learning. However, traditional meta-learning methods rely on explicitly performing learning on tasks drawn from each task distribution they need to adapt to – a costly proposition. By directly learning the mapping between task distributions and initial weights, we can immediately construct artificial neural networks that correspond to particular prior distributions. Our results here show that this approach is possible, characterize the optimal solution in the linear case, and demonstrate that this idea is viable in more complex models. We see this as an important first step towards being able to automate meta-learning to allow us to more efficiently create learners with specific inductive biases.

ETHICS STATEMENT

This paper presents work whose goal is to improve flexibility of meta-learning algorithms. Increasing the power of machine learning methods has the potential for both positive and negative societal consequences, but we do not see this work as having greater risk than other fundamental research in this area. We note that care should be taken when the task distribution parameters go outside the range used for training, as the hypernetwork output may not be reliable in this case.

REPRODUCIBILITY STATEMENT

Assumptions for our theoretical result (Theorem 3.1) are explained in Section 3.2. The corresponding proof is presented in Appendix A. An overview of our experimental setup is presented in the corresponding sections: Sections 4.1-4.3. Further experimental details may be found in Section B. Code to reproduce our experiments may be found in the supplementary material.

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

## A  PROOF OF THEOREM 3.1

In this section we provide proofs for all theoretical claims from the main paper.

**Theorem 3.1** (Optimal model hyperparameters for meta-learned Bayesian linear regression). *Let the task distribution and model distribution be defined as in Section 3.2. Then the model hyperparameters that minimize the negative log marginal likelihood $J(\mathbf{m}_0, \mathbf{P}_0)$ are given by:*

$$\mathbf{m}_0^* = (\mathbf{X}^\top \mathbf{X})^{-1} \mathbf{X}^\top \boldsymbol{\Phi} \boldsymbol{\mu}_\mathbf{w} \tag{11}$$

$$\mathbf{P}_0^* = (\mathbf{X}^\top \mathbf{X})^{-1} \mathbf{X}^\top \left[ \mathbf{r}\mathbf{r}^\top - \left( \boldsymbol{\Phi} \boldsymbol{\Sigma}_\mathbf{w} \boldsymbol{\Phi}^\top + (\sigma_0^2 + \sigma^2)\mathbf{I} \right) \right] \mathbf{X}(\mathbf{X}^\top \mathbf{X})^{-1}, \tag{12}$$

*where* $\mathbf{r} = (\mathbf{X}\mathbf{m}_0^* - \boldsymbol{\Phi}\boldsymbol{\mu}_\mathbf{w})$.

*Proof.* We begin by simplifying $J(\mathbf{m}_0, \mathbf{P}_0)$ using the form of a multivariate Gaussian density.

$$J(\mathbf{m}_0, \mathbf{P}_0) = -\log \mathcal{N}(\mathbf{X}\mathbf{m}_0 \mid \boldsymbol{\Phi}\boldsymbol{\mu}_\mathbf{w}, \mathbf{X}\mathbf{P}_0\mathbf{X}^\top + \boldsymbol{\Phi}\boldsymbol{\Sigma}_\mathbf{w}\boldsymbol{\Phi}^\top + (\sigma_0^2 + \sigma^2)\mathbf{I}) \tag{17}$$

$$= \frac{1}{2}\mathbf{r}^\top \left(\mathbf{X}\mathbf{P}_0\mathbf{X}^\top + \boldsymbol{\Phi}\boldsymbol{\Sigma}_\mathbf{w}\boldsymbol{\Phi}^\top + (\sigma_0^2 + \sigma^2)\mathbf{I}\right)^{-1} \mathbf{r}$$

$$+ \frac{1}{2}\log\left|\mathbf{X}\mathbf{P}_0\mathbf{X}^\top + \boldsymbol{\Phi}\boldsymbol{\Sigma}_\mathbf{w}\boldsymbol{\Phi}^\top + (\sigma_0^2 + \sigma^2)\mathbf{I}\right| + \frac{n}{2}\log(2\pi),$$

where $\mathbf{r} = (\mathbf{X}\mathbf{m}_0 - \boldsymbol{\Phi}\boldsymbol{\mu}_\mathbf{w})$.

**Solving for $\mathbf{m}_0^*$.** In order to find the optimal $\mathbf{m}_0$, we take the gradient with respect to $J(\mathbf{m}_0, \mathbf{X})$ and set it to zero.

$$\nabla_{\mathbf{m}_0} J = \mathbf{X}^\top \left(\mathbf{X}\mathbf{P}_0\mathbf{X}^\top + \boldsymbol{\Phi}\boldsymbol{\Sigma}_\mathbf{w}\boldsymbol{\Phi}^\top + (\sigma_0^2 + \sigma^2)\mathbf{I}\right)^{-1} (\mathbf{X}\mathbf{m}_0 - \boldsymbol{\Phi}\boldsymbol{\mu}_\mathbf{w}) = 0. \tag{18}$$

For this condition to hold in general, we need

$$\mathbf{X}\mathbf{m}_0 - \boldsymbol{\Phi}\boldsymbol{\mu}_\mathbf{w} = 0 \Rightarrow \tag{19}$$

$$\mathbf{m}_0^* = (\mathbf{X}^\top\mathbf{X})^{-1}\mathbf{X}^\top\boldsymbol{\Phi}\boldsymbol{\mu}_\mathbf{w}. \tag{20}$$

**Solving for $\mathbf{P}_0^*$.** For notational convenience, let $\tilde{\boldsymbol{\Sigma}} = \mathbf{X}\mathbf{P}_0\mathbf{X}^\top + \boldsymbol{\Phi}\boldsymbol{\Sigma}_\mathbf{w}\boldsymbol{\Phi}^\top + (\sigma_0^2 + \sigma^2)\mathbf{I}$. Then $J(\mathbf{m}_0, \mathbf{P}_0)$ may be written concisely as:

$$J(\mathbf{m}_0, \mathbf{P}_0) = -\log \mathcal{N}(\mathbf{X}\mathbf{m}_0 \mid \boldsymbol{\Phi}\boldsymbol{\mu}_\mathbf{w}, \tilde{\boldsymbol{\Sigma}}). \tag{21}$$

The partial derivative with respect to $\tilde{\boldsymbol{\Sigma}}$ may be computed using known formulas (Petersen & Pedersen, 2012, Eq. 396):

$$\frac{\partial J}{\partial \tilde{\boldsymbol{\Sigma}}} = \frac{1}{2}\left[\tilde{\boldsymbol{\Sigma}}^{-1} - \tilde{\boldsymbol{\Sigma}}^{-1}\mathbf{r}\mathbf{r}^\top\tilde{\boldsymbol{\Sigma}}^{-1}\right]. \tag{22}$$

Using the definition of $\tilde{\boldsymbol{\Sigma}}$, we can compute the partial derivative of $J$ with respect to $\mathbf{P}_0$ by using the matrix-valued chain rule:

$$\frac{\partial J}{\partial \mathbf{P}_0} = \mathbf{X}^\top \frac{\partial J}{\partial \tilde{\boldsymbol{\Sigma}}}\mathbf{X} = \frac{1}{2}\mathbf{X}^\top\left[\tilde{\boldsymbol{\Sigma}}^{-1} - \tilde{\boldsymbol{\Sigma}}^{-1}\mathbf{r}\mathbf{r}^\top\tilde{\boldsymbol{\Sigma}}^{-1}\right]\mathbf{X}. \tag{23}$$

Therefore, for the partial derivative to be zero in general, we need

$$\tilde{\boldsymbol{\Sigma}}^{-1} = \tilde{\boldsymbol{\Sigma}}^{-1}\mathbf{r}\mathbf{r}^\top\tilde{\boldsymbol{\Sigma}}^{-1}. \tag{24}$$

Left-multiplying by $\tilde{\boldsymbol{\Sigma}}$ implies that $\tilde{\boldsymbol{\Sigma}} = \mathbf{r}\mathbf{r}^\top$. Substituting into the definition of $\tilde{\boldsymbol{\Sigma}}$ and rearranging,

$$\mathbf{X}\mathbf{P}_0\mathbf{X}^\top = \mathbf{r}\mathbf{r}^\top - \left(\boldsymbol{\Phi}\boldsymbol{\Sigma}_\mathbf{w}\boldsymbol{\Phi}^\top + (\sigma_0^2 + \sigma^2)\mathbf{I}\right). \tag{25}$$

Solving for $\mathbf{P}_0$, we find that

$$\mathbf{P}_0^* = (\mathbf{X}^\top\mathbf{X})^{-1}\mathbf{X}^\top\left[\mathbf{r}\mathbf{r}^\top - \left(\boldsymbol{\Phi}\boldsymbol{\Sigma}_\mathbf{w}\boldsymbol{\Phi}^\top + (\sigma_0^2 + \sigma^2)\mathbf{I}\right)\right]\mathbf{X}(\mathbf{X}^\top\mathbf{X})^{-1}. \tag{26}$$

$\square$

# B ADDITIONAL EXPERIMENTAL DETAILS

In this section we provide additional experimental details.

## B.1 HYPERNETWORK TRAINING PROCEDURE

The detailed training procedure for the hypernetworks used in Sections 4.2 and 4.3 is as follows:

- **Input**: a sampled prior mean $\mu_w$
- **Output**: a predicted initialization $\theta_{\text{init}} \in \mathbb{R}^P$ for the learner
- **Training**:
    1. draw task weights $w_i \sim N(m(\mu_w), \sigma_w^2 I_P)$,
    2. sample support/query sets from $f_{w_i}$,
    3. perform one-step adaptation from $\theta_{\text{init}} = h_\phi(\mu_w)$ on the support set to obtain $\theta_i$,
    4. measure the loss on that task's query set.

    We average those query losses into a scalar hyper-loss, and backpropagate **through the inner update** to update the hyper-network parameters $\phi$.

## B.2 SETTINGS FOR LINEAR EXPERIMENT

The settings for the algorithms evaluated in Section 4.2 were as follows:

**Common settings across all cases**
Noise: $\sigma_y = 0.1$
Task weight variance: $\sigma_w = 1.0$
Inner learning rate (adaptation): $\alpha = 0.01$ (one step)
Support/query sizes: $k_{\text{spt}} = 10$, $k_{\text{qry}} = 10$
Evaluation tasks per prior: 50
Patience: 10 validation intervals
Validation interval: every 50 steps

**(1) Hyper-network $h_\phi(\mu) \to \theta_{\text{init}}$**
Initialization: predicted $\theta_{\text{init}}$ from a $2 \to 64 \to 64 \to P$ MLP (ReLU)
Outer optimizer: AdamW (weight decay $1 \times 10^{-4}$)
Hyper learning rate: $1.3 \times 10^{-2}$
Hyper steps: 2000
Batching: 4 priors per batch, 8 tasks per prior
Regularization: $8 \times 10^{-4} \cdot \|\theta_{\text{init}}\|^2$
Objective: minimize average query loss after one inner update (with 3 inner steps, per-parameter $\alpha$)

**(2) Pooled MAML (shared initialization across priors)**
Initialization: one shared $\theta_{\text{init}}$ across all training priors
Outer optimizer: Adam
Meta learning rate: $1 \times 10^{-2}$
Outer steps: 1000
Meta-batch size: 100 tasks per step, sampled from random priors
Objective: minimize query loss averaged over tasks from all priors

**(3) Per-prior MAML**
Initialization: one $\theta_{\text{init}}$ retrained separately for each test prior
Outer optimizer: Adam
Meta learning rate: $1 \times 10^{-2}$
Outer steps: 1000
Meta-batch size: 100 tasks per step, sampled from that prior
Objective: minimize query loss tailored to the given test prior

## B.3 SETTINGS FOR MLP EXPERIMENT

The settings for the algorithms evaluated in Section 4.3 were as follows.

**Common settings across all cases**
Learner model: two-layer MLP ($2 \to 32 \to 1$, LeakyReLU activation with slope 0.1)
Task weight variance: $\sigma_w = 0.3$
Observation noise: $\sigma_y = 0.1$
Inner adaptation: $K = 3$ steps with per-parameter step sizes $\alpha = \text{softplus}(\rho)$, bounded in $[1 \times 10^{-5}, 5 \times 10^{-2}]$
Support/query sizes (train): $k_{\text{spt}} = 10$, $k_{\text{qry}} = 20$
Support/query sizes (eval): $k_{\text{spt}} = 20$, $k_{\text{qry}} = 200$
Evaluation tasks: 16
Task weight prior means: $m(\mu)$ given by frozen linear map $\mathbb{R}^2 \to \mathbb{R}^P$
Optimizer: gradients clipped to norm 0.5
Validation interval: 10 steps; Patience: 15 intervals

**(1) Hyper-network** $h_\phi(\mu) \to \theta_{\textbf{init}}$
Architecture: $2 \to 64 \to 64 \to P$ MLP (ReLU)
Outer optimizer: AdamW (weight decay $1 \times 10^{-4}$)
Hyper learning rate: $1.3 \times 10^{-2}$
Hyper steps: 2000
Batching: 4 priors per batch, 8 tasks per prior
Regularization: $8 \times 10^{-4} \cdot \|\theta_{\text{init}}\|^2$
Objective: minimize query loss after adaptation with shared per-parameter step sizes

**(2) Pooled MAML (shared initialization across priors)**
Initialization: one shared $\theta_{\text{init}}$ across all training priors
Outer optimizer: Adam
Meta learning rate: $1 \times 10^{-2}$
Meta steps: 400
Meta-batch size: 32 tasks per step, sampled across random priors
Objective: minimize query loss averaged over all priors

**(3) Per-prior MAML oracle**
Initialization: one $\theta_{\text{init}}$ retrained separately for each test prior
Outer optimizer: Adam
Meta learning rate: $1 \times 10^{-2}$
Meta steps: 400
Meta-batch size: 32 tasks per step, sampled from that prior
Objective: minimize query loss specific to the test prior (oracle baseline)

