# OpenReview forum: "Automating Meta-learning by Learning to Map Task Distributions to Initial Weights"
_ICLR.cc/2026/Conference — Submitted to ICLR 2026_

### Official Review · Reviewer_3kEY · 2025-10-15

**Soundness:** 3
**Presentation:** 2
**Contribution:** 2
**Rating:** 2
**Confidence:** 4

**Summary:**

The submission proposes to automatize the process of MAML-style meta-learning by learning a mapping from task environment parameters to initial model weights. Formal derivations are presented in a hierarchical Bayesian form where meta-learning happens by marginal likelihood maximization. This is illustrated by the case of Bayesian linear regression, where analytic expressions emerge. In a second, somewhat disconnected part, a hypernetwork-based approach is proposed, namely to directly train a hypernetwork is trained to predict adapted model initializations from task environment parameters. Illustrative experiments are provided with Gaussian distributions in $\mathbb{R}^2$.

**Strengths:**

+ the problem setting of automatizing meta-learning is new
+ the provided derivations seems correct (up to a lack of clarity, see below)

**Weaknesses:**

Unfortunately, there submission has a number of shortcomings.

*Motivation*

The work aims automatizing the process of meta-learning: given many
related meta-learning problems with parametric descriptions, it
learns a hypernetwork to perform the model adaptation instead of
having to do it numerically.
However, I see no real-world scenario where this problem would occur:
while single task environments can occur, e.g. a distribution over
users of an online platform, I cannot imagine a case where many
related such environments would be available for a training step,
and where each environment would have a known parameter vector,
except in simulations.

*Clarity of presentation*

The manuscript has two largely independent parts, the Bayesian
derivation of Section 3.1/3.2 and the Hypernetwork-based approach
in Section 3.3

1) Currently, the Bayesian formulation lacks clarity, presumably because
some key aspects are not explained well enough.

+ the role of $x$: in (3), it appears that all tasks share the same marginal
$p(x)$ and only differ in their conditional $p(y|x)$. That feels like an
strong but unrealistic condition that is not explained. In (4)/(5),
X does not appear anymore. It has proably not been marginalized out,
because $y$ depends on it, so should I read the expressions as implicitly
conditioned on (or a function of) $x$?

+ the dependence of $Y$ on $\Phi$/$X$: (6) allows a nonlinear feature
map for the target function $f_w$. (7) assume the predictive distribution
to be Gaussian centered at a $X\theta$, so $Y$'s dependence on $X$ is
once nonlinear and once linear. Of course, there are different other
quantities involved, but the description left me puzzled about the
intuition of this construction.

A minor issue: $\mathbf{y}$ initially denotes a single output, but later it is used as a vector of outputs.
For $\mathbf{x}$, a different convention ($x_i$, $X$ is used).

Unfortunately, I did not find the three provided references helpful to
justify or clarify the marginal likelihood setup, as they cover quite
different topics.


2) the hypernetwork aspect is quite short and detached from the
Bayesian part. The objective (14) is simply minimizing the Eucidean
distance between model output and target, which does not need a
Bayesian derivation. The argument "Reptile can be viewed as approximately
maximizing marginal likelihood" is also not very powerful.

In fact, I would appreciate a more precise formulation the procedure, e.g. in form of pseudocode, given that plugging (13) into the expression for $\hat\theta$ (line 262) and inserting it into (14) trivialized the expression.
Therefore, something else must go on, such as the arrow in (13) indicating that $\theta_0$ should be treated as constant and not as a function of the hypernetwork. This should be clarified.


*Scientific contribution*

I find the scientific contribution quite limited. The merits of meta-learning
with a marginal likelihood objective is not clear to me, yet, please see my
questions below. The subsequent analysis of the Bayesian linear regression
setting contains little useful insight for me, it feels more like an exercise
in Bayesian inference with Gaussians.
The hypernetwork part suggests a straight-forward way to train a hypernetwork
such that it predicts one vector (the result of reptile) from another vector
(the task parameters). This makes sense in this setup, but I do not consider
it a major conceptual contribution.


*Experimental evaluation*

The experimental evaluation did not convince me. The setup is highly artificial,
using two-dimensional Gaussian data. The first experiments just confirms that the
analytics expression are correct. The other experiments trains a very small
network, still in the linear 2D Gaussian setup. The results Table 3 reports
one numeric experiments (measured in two ways), without error bars or test of
statistical significance.

**Questions:**

* please clarify the notation in the marginal likelihood setup

* what are real-world scenario in which many meta-learning tasks need
  to be solved, and parameter vectors are available as hypernetwork
  inputs?

Please keep your answers short and scientific, then I will be happy to
engage in a discussion.

---

### Official Review · Reviewer_QKgG · 2025-10-22

**Soundness:** 2
**Presentation:** 2
**Contribution:** 2
**Rating:** 2
**Confidence:** 4

**Summary:**

This paper proposes a framework for automating meta-learning by learning a direct mapping from task distribution parameters to Bayes-optimal learner parameters. The method uses hypernetworks to predict the initial weights for a model given the parameters of the task distribution, enabling fast adaptation to new environments without rerunning meta-learning every time.

**Strengths:**

- The paper is well written and easy to follow
- The proposed idea of mapping task distributions to model weights is interesting.
- The theoretical analysis provides an intuitive understanding of the method's foundations.

**Weaknesses:**

- A central claim of the paper is that the proposed approach enables rapid adaptation to new task distributions without retraining. However, this is not convincingly demonstrated in the experiments. An analysis of performance on out-of-distribution (OoD) task distributions is necessary to substantiate this claim.
- The experimental evaluation is limited. Additional baselines—such as Bayesian meta-learning methods, hypernetwork-based meta-learning approaches, and meta-learning algorithms designed for cross-task distribution generalization—should be included for a fair comparison.
- The experiments are restricted to small-scale synthetic tasks. It would strengthen the paper to show applicability to more complex architectures (e.g., CNNs or Transformers) and realistic datasets.
- An analysis of the computational complexity (in terms of memory usage, training time, and computational cost) is missing.
- Reporting standard deviations or confidence intervals for all performance metrics would make the results more robust and comparable.

**Questions:**

- Can the proposed approach adapt to any OoD task, even when the task distribution is very different than the one used for training? In other words, how does the hypernetwork behave when faced with a task distribution far from the training set?
- In Table 2, why does the proposed method outperform the oracle baseline?
- How were the hyperparameters chosen, and were they tuned separately for each baseline?
- Is the method applicable to larger architectures? How does the hypernetwork scale with the dimensionality of the model parameters?
- Does the limited query set size in the experiments affect the reported performance or stability of the results?

---

### Official Review · Reviewer_u7vP · 2025-10-28

**Soundness:** 3
**Presentation:** 2
**Contribution:** 2
**Rating:** 4
**Confidence:** 4

**Summary:**

The paper introduces a framework that learns a direct mapping from the parameters of a task distribution to the initial weights of a model. This allows adaptation to new environments without re-running meta-learning. The paper derives a closed-form Bayes-optimal mapping between task distribution parameters and model hyperparameters for Bayesian linear regression.

**Strengths:**

- Reinterprets meta-learning as a mapping problem from task distributions to model priors. This reframing is conceptually simple yet powerful, potentially transforming how adaptation is handled in meta-learning.
- The proposed hypernetwork offers meta-level amortization: meta-learn once, then generalize to new task distributions with a single forward pass.
- The hierarchical Bayesian perspective is well-articulated, providing strong theoretical continuity with existing literature.

**Weaknesses:**

- The experiments are limited that only toy regression and small MLP tasks are considered. No experiments on high-dimensional or real-world domains (e.g., vision, NLP, RL), making it unclear how the method scales or generalizes. Additionally, the baselines are narrow, including only MAML and one of its variants.
- The authors highlight efficiency as a benefit, yet they do not provide an ablation study on the impact of hypernetwork size on performance, nor do they include a detailed analysis of computational complexity or inference/training time.

**Questions:**

The method relies on explicit access to task distribution parameters $\eta$. When these parameters are unknown or difficult to estimate, how to handle such cases?

---

### Official Review · Reviewer_qfNk · 2025-10-29

**Soundness:** 2
**Presentation:** 3
**Contribution:** 2
**Rating:** 2
**Confidence:** 4

**Summary:**

This paper presents a meta-learning method that uses a hypernetwork to predict model initialization weights, then runs MAML on this initialization to create the final model for a task.  Here, the tasks fall under a hierarchical probability model $p(\mu) -> \mu(w) -> w$ where w is a task and \mu is a distribution over tasks.  The hypernetwork is used to predict a model init $\theta_0$ for MAML given a task distribution $\mu$, which MAML can then use as its init point for adapting to tasks $w$ drawn from $\mu$.

The setup is analyzed in the context of linar-gaussian models where all distributions mentioned above are gaussians with means determined by a linear model of the sample from the layer before.  Numerical experiments are performed on simple examples including the linear-gaussian case and a low-dimensional MLP (dims 2 -> 32 -> 1), finding improvements over constant init learned by vanilla MAML in these small test settings.

**Strengths:**

The use of a hypernetwork to output initializations that are further refined is an interesting construction and strategy.  The paper formulates this setup well and develops its soundness in theoretical simple case (linear-Gaussian).  It has good experiments in small, toy settings that illustrate and confirm its behavior.

**Weaknesses:**

The largest weakness with this paper is it only studies the method in very small, toy settings.  While these confirm that it's basically working as intended, they don't go very far in illustrating its performance.  Likewise, the theoretical results only show the existence and form of a solution in a limited case; while that demonstrates a basic level of soundness, it doesn't seem to say much about the method's performance beyond its ability to converge in a simple case.

In addition, I couldn't find much in the way of concrete descriptions for how $\mu$ might be set up in a more realistic setting.  Table 1 mentions a few applications, but there are no concrete constructions that actually demonstrate how this might work.  Without this, I'm a little bit confused about the motivation for the task distribution $\mu$, and why it's used as the hypernetwork inputs as opposed to the task w directly.  See questions below for more details on these points.

**Questions:**

* Why use $h(\mu)$, as opposed to applying h directly to the task w, as $\theta_0 = h(w)$ to get \theta_0 and run MAML on that?  The w distribution $p(w|\mu)p(\mu|\eta)$ can remain the same as it is now when training the hypernetwork; the only difference is h is applied to w instead of the \mu params.  It would be good to compare to this as well, as it seems a somewhat simpler setup to me.

* What is an example of \mu and w  in a more concrete setting, like image classification?  from what I understand, in this case a task w is a set of classes, and \mu is a distrib over sets of classes.  But for example in benchmarks like mini-/tiered-Imagenet, classes are split between meta-train and meta-test, so that no class in the meta-test (which benchmarks generalization into unknown class deployments) occurs in the meta-training set.  What can \mu be in this case, and how can it be used at test-time?

* formulation with x ~ p(x), y ~ p(y | x, w):  This supposes inputs x are shared between all tasks w.  Which is certainly a valid way to formulate.  But for image classification, for example, if w are the classes and x ~ p(x) from all images over some large image distribution, then it's possible many images don't contain any class in w.  How does this affect the model in practice, perhaps in terms of sampling methods?

* Eq 4:  I don't entirely understand how X and w fit into this objective.  I checked the Snell 21 reference and they have X in their conditioning, but also formulate their objectives in an overall quite different context so it's not immediately clear how to make this correspond.  Since y ~ p(y|x,w), where does p(x) come into play in the eq 4 objective?

* l.355 baselines "Pooled MAML, is equivalent to our method but has no hypernetwork" --- would this be equivalent to vanilla Reptile?  it would be if it's trained on a single distribution of tasks, but not sure if this is the case with sampled task distribs.  OTOH, if tasks are sampled from \mu_w and \mu_w is sampled from a task distrib prior, this implies a modeled distribution on the tasks w as p(\mu) -> \mu -> w, and so pooled maml seems it would be the same as reptile under this overall distrib for w?

---

### Meta-Review · Area_Chair_rnAk · 2025-12-30

**Summary:**

The reviewers agree that while the problem setup of accelerating meta-learning via hypernetworks has some merits, the biggest issue is the lack of convincing examples demonstrating the effectiveness of the method beyond toy problems. The reviewers also the important question of which real-life scenarios would warrant the application of the proposed method. The reviewers also highlighted some clarity issues.

**Reviewer Concerns:**

The authors did not provide a rebuttal.

**Reviewer Scores:**

There would be no change in scores.

---

### Decision · Program_Chairs · 2026-01-26

Reject